

# Predicting groundwater recharge for varying landcover and climate conditions: – a global meta-study

Chinchu Mohan[1], Andrew W. Western[1], Yongping Wei[2] and Margarita Saft[1]

[1] Department of Infrastructure Engineering, University of Melbourne, Melbourne, Victoria, Australia

[2] School of Geography, Planning and Environmental Management, The University of Queensland, Brisbane, Australia

## Abstract

Groundwater recharge is one of the important factors determining the groundwater development potential of an area. Even though recharge plays a key role in controlling groundwater system dynamics, much uncertainty remains regarding the relationships between groundwater recharge and its governing factors at a large scale. The aims of this study were to identify the most influential factors on groundwater recharge, and to develop an empirical model to estimate diffuse rainfall recharge at a global-scale. Recharge estimates reported in the literature from various parts of the world (715 sites) were compiled and used in model building and testing exercises. Unlike conventional recharge estimates from water balance, this study used a multimodel inference approach and information theory to explain the relation between groundwater recharge and influential factors, and to predict groundwater recharge at $0.5^0$ resolution. The results show that meteorological factors (precipitation and potential evapotranspiration) and vegetation factors (land use and land cover) had the most predictive power for recharge. According to the model, long term global average annual recharge (1981-2014) was 134 mm/yr with a prediction error ranging from -8 mm/yr to 10 mm/yr for 97.2% of cases. The recharge estimates presented in this study are unique and more reliable than the existing global groundwater recharge estimates because of the extensive validation carried out using both independent local estimates collated from the literature and national statistics from Food and Agriculture Organisation (FAO). In a water scarce future driven by increased anthropogenic development, the results from this study will aid in making informed decision about groundwater potential at a large scale.

**Keywords**: *Global groundwater recharge, multimodel inference approach, meta study*

## 1 Introduction

Human intervention has dramatically transformed the planet's surface by altering land use and land cover and consequently the hydrology associated with it. In the last 100 years the world population has quadrupled, from 1.7 billion (in 1900) to more than 7.3 billion (in 2014), and is expected to continue to grow significantly in the future (Gerland et al., 2014). During the last century, rapid population growth and the associated shift to a greater proportion of irrigated food production, led to an increase in water extraction by a factor of ~6. This eventually resulted in the over exploitation of both surface and groundwater resources, including the depletion of 21 of the world's 37 major aquifers (Richey et al., 2015). This depletion threatened human lives in many ways, ranging from critical reductions in water availability to natural



disasters such as land subsidence (Chaussard et al., 2014;Ortiz‑Zamora and Ortega‑
Guerrero, 2010;Phien-Wej et al., 2006;Sreng et al., 2009). Therefore, there is a need to closely
examine approaches for sustainably managing this resource by carefully controlling
withdrawal from the system.
Groundwater recharge is one of the most important limiting factors for groundwater withdrawal
and determines the groundwater development potential of an area (Döll and Flörke, 2005)
Groundwater recharge connects atmospheric, surface and subsurface components of the water
balance and is sensitive to both climatic and anthropogenic factors (Gurdak, 2008;Herrera‑
Pantoja and Hiscock, 2008;Holman et al., 2009;Jyrkama and Sykes, 2007). Various studies
have employed different methods to estimate groundwater recharge including tracer methods,
water table fluctuation methods, lysimeter methods, and simple water balance techniques.
Some of these studies input recharge to numerical groundwater models or dynamically link it
to hydrological models to estimate variations under different climate and land cover conditions
(Aguilera and Murillo, 2009;Ali et al., 2012;Herrera‑Pantoja and Hiscock, 2008;Sanford,
58    2002).

In the last few decades, interest in global-scale recharge analysis has increased for various
scientific and political reasons (Tögl, 2010). L′vovich (1979) made the first attempt at a global-
scale by creating a global recharge map using baseflow derived from river discharge
hydrographs. The next large scale groundwater recharge estimate was done by Döll (2002) who
modelled global groundwater recharge at a spatial resolution of $0.5^0$ using the WaterGAP
Global Hydrological model (WGHM) (Alcamo et al., 2003;Döll, 2002). In this study, the
runoff was divided into fast surface runoff, slow subsurface runoff and recharge using a
heuristic approach. This approach considered relief, soil texture, hydrogeology and occurrence
of permafrost and glaciers for the runoff partitioning. However, WGHM failed to reliably
estimate recharge in semi-arid regions (Döll, 2002). Importantly, in that study, there was no
consideration of the influence of vegetation which has been reported to be the second most
important determinant of recharge by many researchers (Jackson et al., 2001;Kim and Jackson,
2012;Scanlon et al., 2005). In subsequent years, several researchers have attempted to model
global groundwater recharge using different global hydrological models and global-scale land
surface models (Koirala et al., 2012;Scanlon et al., 2006;Wada et al., 2010).
Although a fair amount of research has been carried out to model groundwater recharge at a
global-scale, most studies compared results to country level groundwater information from the
FAO (FAO, 2005). The inconsistent and approximate nature of FAO estimates raises questions
about the reliability of its use as a standard comparison measure. No study has validated
modelled estimates against small scale recharge measurements. In addition, research has been
mostly restricted to studying meteorological influences on recharge, few studies have
systematically explored global-scale factors governing recharge. Much uncertainty still exists
about the relationship between groundwater recharge and topographical, lithological and
vegetation factors. Without adequate knowledge of these controlling factors, our capacity to
sustainably manage groundwater globally will be seriously compromised.
The major objectives of this study are to identify the most influential factors on groundwater
recharge and to develop an empirical model to estimate diffuse rainfall recharge. Specifically,





to quantify regional effects of meteorological, topographical, lithological and vegetation
factors on groundwater recharge using data from 715 globally distributed sites. These
relationships are used to build an empirical groundwater recharge model and then the global
groundwater recharge is modelled at a spatial resolution of $0.5^0$ x $0.5^0$ for the time period 1981
– 2014.
**2    Methods**
2.1    Dataset
This study is based on a compilation of recharge estimates reported in the literature from
various parts of the world. This dataset is an expansion of previously collated sets of recharge
studies along with the addition of new recharge estimates (Döll and Flörke, 2005;Edmunds et
al., 1991;Scanlon et al., 2006;Tögl, 2010;Wang et al., 2010). The literature search was carried
out using Google scholar, Scopus and Web of science with related keywords 'groundwater
recharge', 'deep percolation', 'diffuse recharge' and 'vertical groundwater flux'. Several
criteria were considered in including each study.  To ensure that the data reflects all seasons,
recharge estimates for time periods less than one year were excluded. The sites with significant
contribution to groundwater from streams or by any artificial means were also eliminated as
the scope of this research was to model naturally occurring recharge. In order to maximize the
realistic nature of the dataset, all studies using some kind of recharge modelling were removed
from the dataset. After all exclusions, 715 data points spread across the globe (Figure 1)
remained and were used for further analysis. Of these studies, 345 were estimated using the
tracer method, 123 using the water balance method, and the remaining studies used baseflow
method, lysimeter, or water table fluctuation method. This diversity in recharge estimation has
enabled us to evaluate systematic differences in various measurement techniques. The year of
measurement or estimation of recharge estimates in the final dataset differed (provided as
supplementary material), and ranged from 1981 to 2014 (Figure 2).  This inconsistency in the
data raised a challenge when choosing the timeframe for factors in the modelling exercise,
particularly those showing inter annual variation.

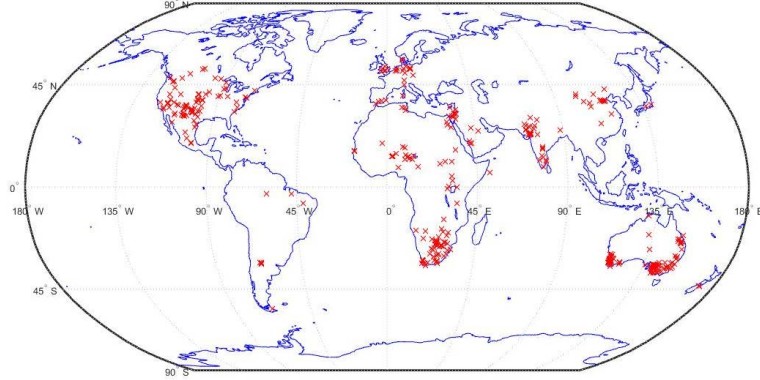

Figure. 1. Locations of the 715 selected recharge estimation sites used for model building.



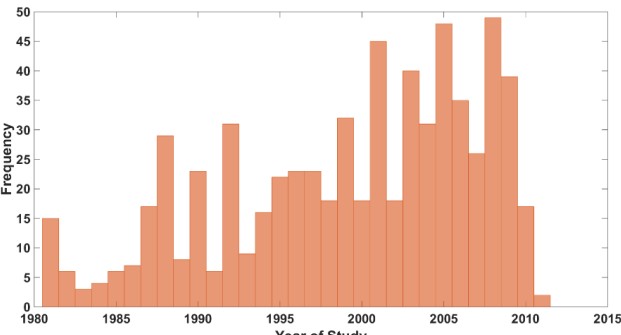


Figure 2. Histogram showing frequency and spread of year of study of recharge estimates in
the final dataset.


The next step was to identify potential explanatory factors that could influence recharge
(referred to as predictors from here on). Potential predictors that were reported in the literature
as having some influence on recharge were identified (Athavale et al., 1980;Bredenkamp,
1988;Edmunds et al., 1991;Kurylyk et al., 2014;Nulsen and Baxter, 1987;O'Connell et al.,
1995;Pangle et al., 2014). The choice of predictors was made based on the availability of global
gridded datasets and relative importance in a physical sense. Finally, we employed 12
predictors comprising meteorological factors, soil/vadose zone factors, vegetation factors and
topographic factors. Details of predictors are given in Table 1.

Data for the chosen predictors corresponding to 715 recharge study sites were extracted from
global datasets. Meteorological datasets ($P$, $T$ and $PET$) were obtained from the Climatic
Research Unit, University of East Anglia, England. Even though daily data was available from
1901 to 2014 at a resolution of $0.5^0$ x $0.5^0$, in this study mean annual average of the latest 34
years (1981 to 2014) was used to reduce the inconsistency in year of recharge measurements
in the final dataset. Topographic and soil data were acquired from the NASA Earth observation
dataset. Both datasets were of $0.5^0$ x $0.5^0$ spatial resolution. A few of the predictors, including
number of rainfall days ($Rd$) and land use/land cover ($LU$) data were obtained from AquaMaps
(by FAO) and USGS (United States Geological Survey) at a spatial resolution of $0.5^0$ x $0.5^0$
and 15 arc minutes respectively. Thus obtained $LU$ data was compared with land cover reported
in literature and corrected for any discrepancies. The spatial resolution of the different data
used was diverse. This was dealt with, by extracting the values for each recharge site from the
original grids using the nearest neighbour interpolation method. As a result, predictor data
extracted for each recharge site will differ from the actual value due to scaling and interpolation
errors. Out of the 12 predictors $LU$ was not a quantitative predictor and was transformed into
a categorical variable in the modelling exercise.

Table 1. Description of predictors used for recharge model building

| Predictors | Symbol | Unit | Resolution | Temporal span | Source | Description | Reference |
|------------|--------|------|------------|---------------|--------|-------------|-----------|
|            |        |      |            |               |        |             |           |





| | | | | | | | |
|---|---|---|---|---|---|---|---|
| Precipitation | $P$ | mm/yr | $0.5^0$ x $0.5^0$ | 1981 - 2014 | Climatic Research Unit, University of East Anglia, England | Mean annual | (Harris et al., 2014) |
| Mean temperature | $T$ | $^0$C | $0.5^0$ x $0.5^0$ | 1981 - 2014 | Climatic Research Unit, University of East Anglia, England | Mean annual temperature | (Harris et al., 2014) |
| Potential evapo-transpiration | $PET$ | mm/yr | $0.5^0$ x $0.5^0$ | 1981 - 2014 | Climatic Research Unit, University of East Anglia, England | Penman-Monteith Reference Crop Evapotranspiration | (Harris et al., 2014) |
| No. of rainy days | $Rd$ | | 5 arc minute | 1981 - 2014 | AQUAM APS, FAO | Average number of wet days per year defined as having $\geq$ 0.1 mm of precipitation | (New et al., 2002) |
| Slope | $S$ | fraction | $0.5^0$ x $0.5^0$ | - | Earth data, NASA | Mean Surface slope | (Verdin, 2011) |
| Saturated hydraulic conductivity | $k_{sat}$ | cm/d | $1^0$ x $1^0$ | - | Earth data, NASA | Saturated hydraulic conductivity at 0 - 150 cm depth | (Webb et al., 2000) |
| Soil Water Storage Capacity | $SWSC$ | mm | $1^0$ x $1^0$ | - | Earth data, NASA | Texture derived soil water storage capacity in soil profile (upto 15 m depth) | (Webb et al., 2000) |
| Excess water (without irrigation) | $EW$ | mm | - | 1981 - 2014 | - | $\sum_{i=1}^{12}(P_i - PET_i)$ where $P_i > PET_i$ | |
| Aridity index | $AI$ | - | - | 1981 - 2014 | - | AI = P/PET | |
| Clay Content | $Clay$ | % | $1^0$ x $1^0$ | - | Earth data, NASA | 0-150cm profile | (DAAC, 2016) |





| Bulk Density | $\rho_b$ | gm/cm$^3$ | $1^0$ x $1^0$ | - | Earth data, NASA | 0-150cm profile | (DAAC, 2016) |
|---|---|---|---|---|---|---|---|
| Land use land cover | $LU$ | - | 15 arc second | - | USGS/Literature | Forest, Pasture, Cropland, Urban/build up, Barren | (Kim and Jackson, 2012;Broxton et al., 2014) |

2.2   Recharge model development
With empirical studies, the science world is always sceptical about whether to use a single best-
fit model or to infer results from several better predicting and plausible models. The former
option is feasible only if there exists a model which clearly surpasses other models, which is
rare in the case of complex systems like groundwater. Usually cross correlation and multiple
controlling influences on the system lead to more than one model having a similarly good fit
to the observations. Thus choosing explanatory variables and model structure is a significant
challenge. In the past this challenge was often addressed using various step-wise model
construction methods, with the final model being selected based on some model fit criteria that
penalises model complexity or results in high numbers of explanatory variables (Fenicia et al.,
2008;Gaganis and Smith, 2001;Jothityangkoon et al., 2001;Sivapalan et al., 2003). These
approaches were pragmatic responses to the large computational load involved in trying all
possible models but they have a disadvantage in that the final model will be dependent on the
step-wise selection process used (Sivapalan et al., 2003). An alternative approach for
addressing this high level of uncertainty in model structure is to adopt a multi-model inference
approach that compares many models (Duan et al., 2007;Poeter and Anderson, 2005). It
typically results in multiple final models and an assessment of the importance of each
explanatory variable. Therefore, this approach was used to develop an understanding of the
role of different controlling factors on recharge in a data limited condition.
Choosing predictors that are capable of representing the system and selecting the right models
for prediction are the key steps in the multi-model inference approach. Here, models were
chosen by ranking the fitted models based on performance, and comparing this to the best
performing model in the set (Anderson and Burnham, 2004). This model ranking also provided
a basis for selecting individual predictors. The analysis progressed through three key stages:
exploratory analysis; model building and model testing.
2.2.1   Multi-model analysis
A multi-model selection process aims to explore a wide range of model structures and to assess
the predictive power of different models in comparison with others. Essentially, models with
all possible combinations of selected predictors are developed and assessed via traditional
model performance metrics (discussed later). By conducting such an exhaustive search, multi-
model analysis avoids the problems associated with selection methods in step-wise regression
approaches (Burnham and Anderson, 2003). Importantly, it reduces the chance of missing
combinations of predictors with good predictive performance. However, a disadvantage of this
approach is that the number of predictor combinations grows rapidly with the number of factors
considered. To make the analysis computationally efficient, we set an upper limit for the
number of predictors used. Another problem with this approach is that it can result in over





fitting. To address this issue we evaluated model performance with metrics that penalise
complexity and tested the model robustness with a cross-validation analysis.   The model
development procedure using multi-model analysis is described in detail below.
(a) Exploratory Analysis
Firstly, all the chosen predictors were individually regressed against the compiled recharge
dataset. This was carried out with the main objective to find the predictors having significant
control on recharge and to gain an initial appreciation of how influential each predictor is
compared to others. This understanding will aid in eliminating the least influencial predictors
from further analysis. Then assumptions involved in regression analysis, such as linearity, low
multicollinearity (important for later multivariate fitting), and independent identically
distributed residuals were analysed using residual analysis. Following the residual analysis,
various data transformations (square root, logarithmic and reciprocal) were carried out to
reduce heteroscedasticity and improve linearity of the variables. The square root transformed
recharge along with non-transformed predictors gave the most homoscedastic relations (results
not shown). Therefore, these transformed values were used in further model building exercises.
Predictors were selected and eliminated based on statistical indicators such as adjusted
coefficient of determination ($R^2_{adj}$) value and Root mean square error (RMSE).
(b) Model building
Multiple linear regression was employed for building the models as the transformed dataset did
not exhibit any nonlinearity. Furthermore, the presence of both negative and positive values in
the dataset restricted the applicability of other forms of regression like log-linear and
exponential (Saft et al., 2016). Linear regression is known for its simple and robust nature in
comparison to higher order analysis. The robustness of linear regression helped to maintain
parsimony together with reasonable prediction accuracy. A rigorous model building approach
was adopted in order to capture the interplay between predictors with combined/interactive
effects on groundwater recharge. This is an exhaustive search in which all candidate models
are fitted and intercompared using performance criteria. In a way, this modelling exercise used
a top-down approach, starting with a simple model which is expanded as shortcomings are
identified (Fenicia et al., 2008).
(c) Model testing
The analysis above provided insight into the relative performance of the models. However, it
is also important to assess the dependence of the results on the particular sample, so we
conducted a subsample analysis in which the same method was re-applied to subsamples of the
data. Finally, predictive uncertainty was estimated through leave-one-out cross validation. In
the first case, the whole model development process was redone multiple times using
subsamples of the data. To achieve this, the entire dataset was randomly divided into 80% and
20% subsets and 80% of the data were used for building the model. The predictive performance
developed model was tested against the omitted 20% of data. This was repeated 200 times, in
order to eliminate random sampling error. The leave-one-out cross validation was applied to
the best few individual model structures and provided an estimate of predictive performance
for those particular models. It also gave an indication of data quality at each point.





In summary the key steps in the multi-model analysis were:

1. Selecting predictors
2. Fitting all possible models consisting of combinations of predictors
3. Determining the optimum number of predictors for each model, $V_{opt}$
4. Calculating model performance metrics for each model up to $V_{opt}$,
5. Calculating the "weight of evidence" for each predictor based on the performance
metric of all models containing that predictor
6. Testing the predictive performance of the models.

2.2.2    Ranking models and predictors
Model performance was evaluated using several information criteria.  These information
criteria include a goodness of fit term and an overfitting penalty based on the number of
predictors in the particular model.   In this study we used $R^2_{adj}$, the Consistent Akaike
Information Criterion (AICc), and the Complete Akaike Information Criterion (CAIC) as the
performance evaluation criteria. These criteria differ in terms of penalising overfitting. $R^2_{adj}$
penalises over-fitting the least, AICc moderately, and CAIC heavily. However, when we are
unsure of the true model and whether it over fits or not, there is some advantage in employing
several criteria as it gives insight into how the results depend on the criteria used.  Suitability
of the information criteria also varies with the sample size. CAIC acts as an unbiased estimator
for large sample size with relatively small candidate models, but produces large negative bias
in other cases, whereas AICc is well suited for small-sample applications (Cavanaugh and
Shumway, 1997;Hurvich and Tsai, 1989). The formulas for the above criteria are as follows:

$AIC = -2 \times llf + 2 \times k$  *(Akaike, 1974)*                    [1]
$AICc = AIC + (2 \times (k-1) \times \frac{k+2}{n-k-2})$  *(Hurvich and Tsai, 1989)*    [2]
$CAIC = -2 \times llf + k \times (ln(n) + 1)$  *(Bozdogan, 1987)*        [3]
$R^2 = 1 - \left[\frac{n-1}{n-k-1}\right] \times [1 - R^2]$  *(Ezekiel, 1929;Wang and Thompson, 2007)*    [4]
where $llf$ is the log-likelihood function, $k$ is the dimension of the model, and $n$ is the number
of observations.

When assessing candidate models there are two aspects which are of particular interest: (1)
which models are better? and (2) how much evidence exists for each of the predictors in
predicting recharge? Analysis of the AICc and CAIC was used to answer both these questions.
Models were ranked using information criteria, with smaller values indicating better
performance. Information criteria are more meaningful when they are used to evaluate the
relative performance of the models (Poeter and Anderson, 2005). Models were ranked from
best to worst by calculating model delta values (Δ) and model weights (*W*) as follows:

$\Delta_i = AIC_i - AIC_{min}$                                [5]
$W_i = exp(-0.5 \times \Delta_i) / \Sigma \, exp(-0.5 \times \Delta_m)$    [6]

where, $AIC_{min}$ is the information criteria value of the best model. $\Delta_i$ and $W_i$ represent the
performance of $i^{th}$ model in comparison with the best performing model in the set of *M* models.





Given that these are relative measures, they are independent of the size of the sample or number
of candidate models.
Evidence ratios were then calculated as the ratio of the $i^{th}$ model weight to the best model
weight. The evidence ratio can be used as a measure of the evidence for the $i^{th}$ model compared
to the other models. The evidence ratios also provide means to estimate the importance of each
predictor. This involves transformation of evidence ratios into a Proportion of evidence (PoE)
for each predictor. PoE for a predictor is defined as the sum of weights of all the models
containing that particular predictor. PoE ranges from 0 to 1. The closer the PoE of a predictor
is to 1, the more influential that predictor is.
2.3 Global groundwater recharge estimation
The best model from the above analysis was used to build a global recharge map at a spatial
resolution of $0.5^0$ x $0.5^0$. Recharge estimation was done annually for a study period of 34 years
(1981–2014), and the estimated groundwater recharge was averaged over the period to produce
a global map. In addition to this, maps showing percentage of rainfall becoming recharge, and
variation of recharge over the years were also generated. As recharge data from regions with
frozen soil were scarce in the model building dataset, the model predictions in those regions
particularly for regions with Koopan classification Dfc, Dfd, ET and EF are not highly reliable,
so the EF regions of Greenland and Antarctica were excluded due to lack of data. However,
the modelled recharge for Dfc, Dfd and ET regions were included in the final map. In addition,
the modelled recharge values were compared against country level statistics from FAO (2005)
for 153 countries.
**3 Results**
The results address three important questions. 1. Which are the most influential predictors of
groundwater recharge? 2. What are the better models for predicting recharge? 3. How does
groundwater recharge vary over space and time? The first question was answered by carrying
out an exploratory data analysis and also by estimating the PoE for each predictor, the second
using information criteria and the third by mapping recharge at $0.5^0$ x $0.5^0$ using the best model.
3.1 Exploratory data analysis
Table 2 gives the statistical summary of predictors and groundwater recharge at 715 data sites.
It is apparent from the table that predictors varied considerably between sites, consistent with
inter-site variability in regional physical characteristics. This variability provided an
opportunity to explore recharge mechanisms in a range of different physical environments. As
we used linear regression to study the one to one relationship of recharge with each of the
predictors, RMSE and bias of fitting were used to identify the predictors with the most
explanatory power. In this case, RMSE values ranged between 23.2 mm/yr for *P* and 30.21
mm/yr for *S*. Predictive potential of meteorological predictors was greater than for other classes
of predictor. (Figure 3). *P, AI, EW* and $\rho_b$ had a negative bias whereas, all other predictors had
a positive bias.
Table 2. Summary statistics of potential predictors from the dataset used in this study.

| Parameters | Minimum | Maximum | Range | Mean | Standard deviation |
|---|---|---|---|---|---|





| | | | | | |
|---|---|---|---|---|---|
| $P$ (mm/yr) | 1.30 | 2627.00 | 2625.70 | 572.82 | 305.65 |
| $T$ ($^0$C) | 1.60 | 30.62 | 29.02 | 17.73 | 6.04 |
| $PET$ (mm/yr) | 6.60 | 2600.00 | 2593.40 | 1356.17 | 401.77 |
| $Rd$ (d/y) | 2.00 | 270.00 | 268.00 | 85.89 | 42.78 |
| $S$ | 0.00 | 10.16 | 10.15 | 0.84 | 1.17 |
| $k_{sat}$ (cm/d) | 0.00 | 265.75 | 265.75 | 60.61 | 59.50 |
| $SWSC$ (mm) | 2.00 | 1121.00 | 1119.00 | 517.38 | 240.81 |
| $AI$ | 0.00 | 68.18 | 68.18 | 0.70 | 3.74 |
| $EW$ (mm/yr) | 0.01 | 1467.87 | 1467.86 | 125.41 | 188.07 |
| $\rho_b$ (gm/cm$^3$) | 0.15 | 1.67 | 1.51 | 1.44 | 0.20 |
| $Clay$ (%) | 1.87 | 52.51 | 50.64 | 23.77 | 7.66 |
| $LU$ | 1.00 | 5.00 | 4.00 | 2.58 | 0.81 |
| $Recharge$ (mm/yr) | 0.00 | 1375.00 | 1375.00 | 73.22 | 125.94 |


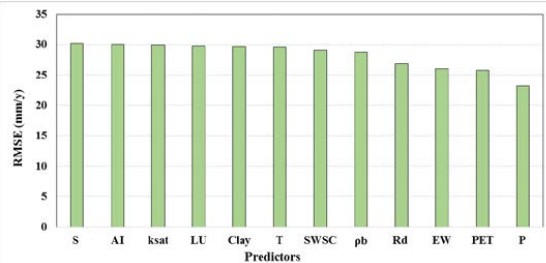

Figure 3. Model fit performance criteria for single predictor regressions.

3.2    Multi-model analysis
3.2.1    Proportion of evidence (PoE) for individual predictors
Figure 4 shows the PoE of the 12 predictors used in this study. According to this analysis, 3 of
the 12 predictors stood out as having the greatest explanatory power (Figure 4). Precipitation
($P$), Potential evapotranspiration ($PET$) and Land use land cover ($LU$) had the highest
proportions of evidence (~1). Subsurface percentage of clay ($Clay$) and Saturated hydraulic
conductivity ($k_{sat}$) also had an important influence on recharge with PoE ~0.4. Aridity index
($AI$), Rainfall days ($Rd$), Mean temperature ($T$), Bulk density ($\rho_d$), Slope ($S$), Excess water
($EW$) and Soil water storage capacity at root zone ($SWSC$) were in the lower PoE range (<0.1
according to both the criteria).  There was some variation in the PoE value of the predictors
with performance metric, due to the diversity in over-fitting penalty. However, ranking of the
variables was identical irrespective of the performance metric used. The 'best' and 'worst'
predictors ranked according to $R^2_{adj}$ were also in agreement with the PoE analysis (not shown).
In addition, results of the subsample analysis gave similar results (not shown).





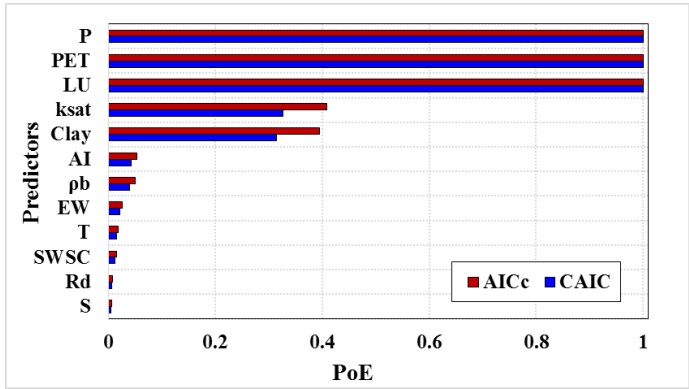

Figure 4. Proportion of evidence according to AICc and CAIC for 12 predictors (sorted in
descending order of PoE).

3.2.2  Better performing models
According to information criteria, the performance of models can only be evaluated relative to
the best performing model in the set. In this study, as per the model weights, no model exhibited
apparent dominance. The evidence ratio (ratio between the weights of the best model and $n^{th}$
model) suggested that the best model according to CAIC was only 1.04 times better than the
2nd best model. However, the evidence ratio increased exponentially with increase in model
rank and there was a clear distinction between better models and worse models. Similar results
were reported by Saft et al. (2016) in her work for modelling rainfall-runoff relationship shift.
The choice of better models was made by considering the PoE of individual predictors (refer
section 3.2.1) and the optimal number of predictors in the model ($V_{opt}$). $V_{opt}$ was chosen by
comparing the performance of the top 10 models out of all possible models that could be
developed with different maximum number of predictors ($V_{max}$). Figure 5 shows the
performance criteria for the top three models for different $V_{max}$ values. The model performance
increased with $V_{max}$ up to 4 or 5, depending on the different criteria. After that, AICc, CAIC
and $R^2_{adj}$ values remained constant, indicating that further addition of predictors did not
improve the model performance. Table 3 illustrates the predictors in the top 10 models
according to performance criteria. *P*, *PET* and *LU* repeatedly appeared in the predictor list of
the top ten models substantiating their high predictive capacity. In this particular case, top
performing models according to both information criteria were the same, therefore results from
only one criteria (CAIC) will be discussed.

Table 3. Predictors used in the top 10 models, ranked based on CAIC criteria (* indicates the
predictor was included).

| Model ranking | P | T | PET | Rd | S | $k_{sat}$ | SWSC | AI | EW | $\rho_b$ | Clay | LU |
|---|---|---|---|---|---|---|---|---|---|---|---|---|
| 1 | * | | * | | | * | | | | | | * |
| 2 | * | | * | | | | | | | | * | * |
| 3 | * | | * | | | | | | | | | * |





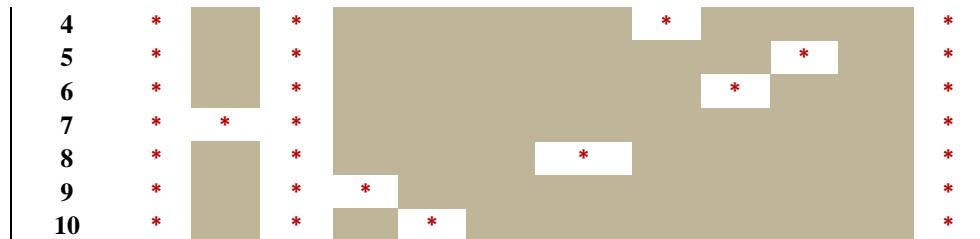



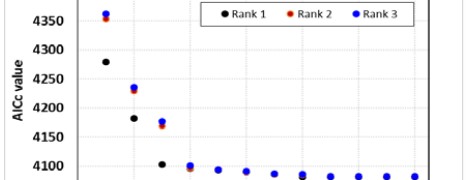 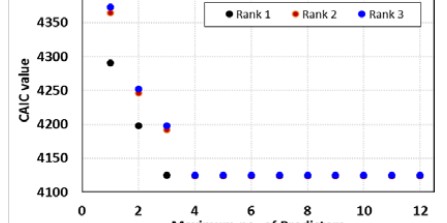

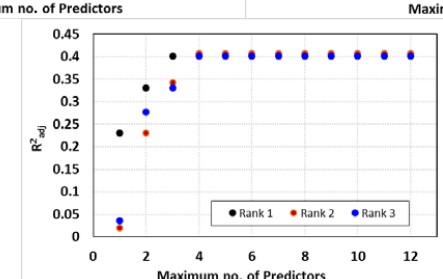

Figure 5. AICc, CAIC, and $R^2_{adj}$ for the top 3 models with varying complexity (number of
predictors, $V_{max}$).

3.2.3    Model testing
Models ranking from 1 to 10 according to CAIC (Table 3) were tested using both the model
testing techniques discussed in section 2.2.1(c). Figure 6 depicts model fit and model prediction
RMSE values of 200 subsample tests. It is clear from the boxplots that the difference between
the RMSE of the 1st and the 10th model during both model fitting and prediction is less than 1
mm/yr. In subsample tests, $R^2_{adj}$ of the best model ranged from 0.42 to 0.56 implying 42 to 56%
of the variance was explained. The model errors at each data point ranged from -8 to 28 mm/yr.
However, 97.2% of the points had errors between -8 and 10 mm/yr. Figure 7 shows the relation
between precipitation and model errors and it is evident from this scatter plot that model
predictions were not greatly influenced by low or high precipitation. In other words, the model
was unbiased by precipitation trends. Similar checking was done for all other predictors (not
shown) which all showed a similar pattern to precipitation. The dataset was classified based on
recharge estimation techniques and model performance was tested with results showing no
systematic difference (not shown).



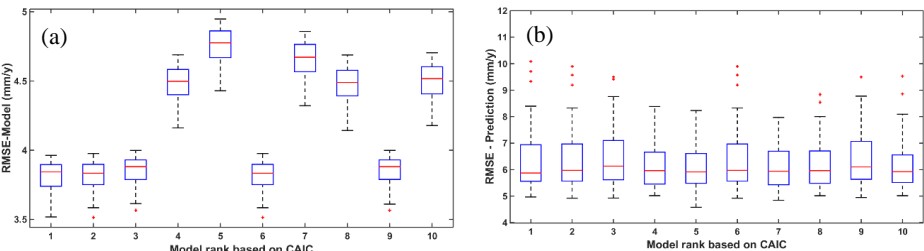


Figure 6. RMSE of sub-sample (a) model fitting and (b) model prediction of top 10 models
according to CAIC.


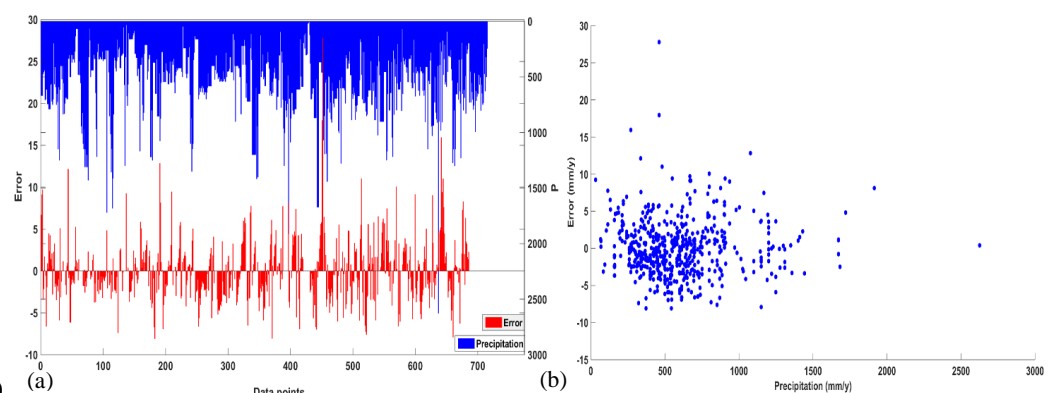


Figure 7 (a) Error at each data point along with the corresponding rainfall obtained using the
leave-one-out model testing procedure and (b) Scatter plot between error at each data point
and corresponding precipitation.

3.3     Global Groundwater Recharge
The global long term (1981 – 2014) mean annual groundwater recharge map at a spatial
resolution of $0.5^0$ was made by the model developed in section 3.2 (Figure 8). Grid scale
recharge ranged from 0.02 mm/yr to 996.55 mm/yr with an average of 133.76 mm/yr. The
highest recharge was associated with very high rainfall (>4000 mm/yr). Humid regions such
as Indonesia, Philippines, Malaysia, Papua New Guinea, Amazon, Western Africa, Chile,
Japan and Norway had very high recharge (>450 mm/yr). Whereas, arid regions of Australia,
the Middle East and Sahara had very low recharge (<0.1 mm/yr). In humid areas, percentage
of rainfall becoming groundwater recharge (>40%) was found to be very high in comparison
to other parts of the world. However, the mean percentage of rainfall becoming recharge is
only 22.06% across the globe. Among all the continents, Australia had the lowest annual
groundwater recharge rate.

Over the 34 years, global annual mean recharge followed the same pattern as that of global
annual mean precipitation (Figure 9). Least recharge was predicted in the year 1987
(groundwater recharge=95 mm/yr), where the annual average rainfall was <180 mm/yr.
Variation in recharge over the years was maximal in arid regions of Australia and North Africa
(Figure 10(a)). However, the standard deviation of recharge was higher in humid areas than in




arid regions (Figure 10(b)). This indicates that standard deviation did not clearly represent year
to year variations in recharge. Potentially, the advantage of using coefficient of variation over
standard deviation is that it can capture variations even when mean values are very small. In
this case precipitation and potential evapotranspiration were the two major predictors of
recharge. Globally, variability in evapotranspiration is much less than variability in rainfall
(Peel et al., 2001; Trenberth and Guillemot, 1995). Therefore, variability of groundwater
recharge both temporally and spatially is due to variability in precipitation, which implies that
arid regions are more susceptible to inter-annual variation in groundwater recharge. A
comparison of predicted recharge against country level recharge estimates from FAO (2005)
shows that the model tends to over predict recharge, particularly for low recharge areas.
However, due to inaccuracies in the FAO estimates this cannot be considered as a reliable
comparison (Figure 11).

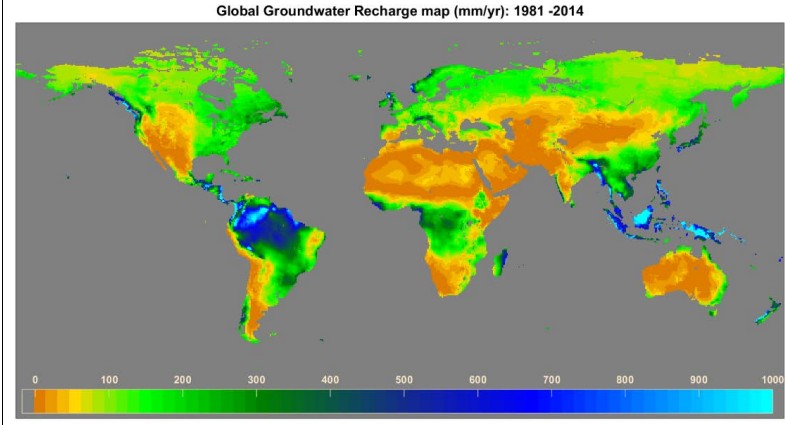


Figure 8. Long-term (1981 -2014) average annual groundwater recharge estimated using the
developed model.

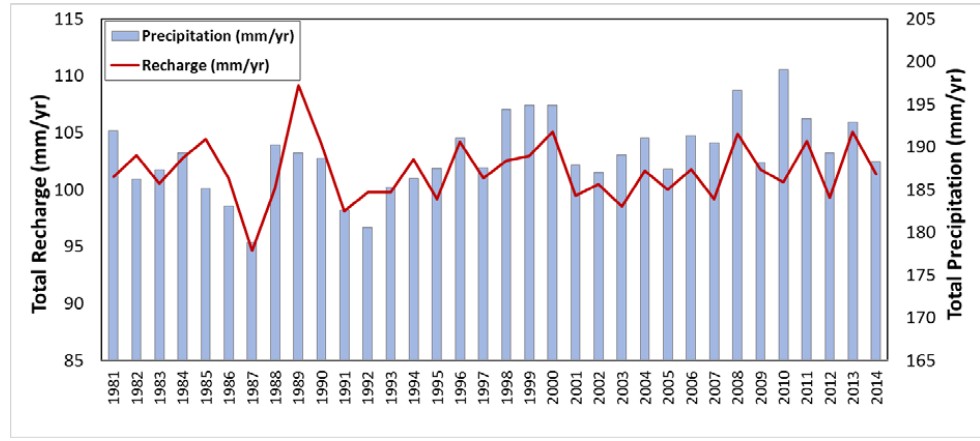


Figure 9. Temporal distribution of total global recharge along with total global precipitation
of corresponding years for a period of 1981 to 2014.






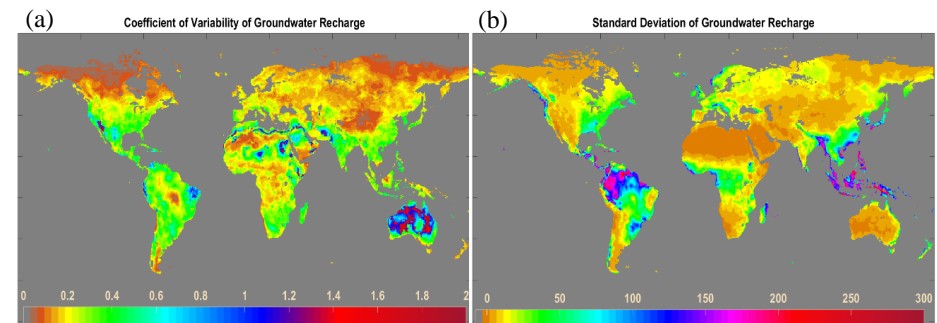


Figure 10. Map showing (a) coefficient of variability and (b) standard deviation of annual
groundwater recharge from 1981 to 2014.

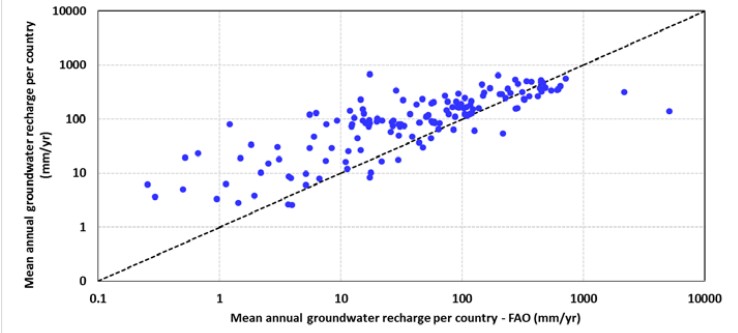


Figure 11. Comparison of predicted recharge against country level estimates from FAO.

**4    Discussion**
The aims of this study were to identify the factors having the most influence on groundwater
recharge, and to develop a global model for predicting groundwater recharge under limited data
conditions, without extensive water balancing. In this study, an empirical model building
exercise employing linear regression analysis, multimodel inference techniques and
information criteria was used to identify the most influential predictors of groundwater
recharge and use them to build predictive models.  Finally, a global groundwater recharge map
was created using the developed model. The key findings from this study and their implications
for future research and practice with respect to global groundwater recharge are discussed
below.

One of the findings to emerge is that, out of numerous models developed in this study there
was no single best model for groundwater recharge.  Instead, there were clear sets of better and
worse models. However, there were predictors which stood out as having greater explanatory
power.  Of the 12 predictors chosen for the analysis, meteorological ($P, PET$) and vegetation
predictors ($LU$) had the most explanatory information followed by saturated hydraulic
conductivity and clay content. Thus models using these predictors ranked higher according to
information criteria. It is reasonable that meteorological factors had the most explanatory
information. In most cases, especially dry regions, groundwater recharge is controlled by the





availability of water at the surface, which is mainly controlled by precipitation,
evapotranspiration and geomorphic features (Scanlon et al., 2002). Numerous studies agree
with this finding. For example, in south western USA, 80% of recharge variation is explained
by mean annual precipitation (Keese et al., 2005). However, the influence of meteorological
factors on groundwater recharge is highly site-specific (Döll and Flörke, 2005). The effect of
meteorological factors can also depend on whether the season or year is wet or dry, type of
aquifer and irrigation intensity (Adegoke et al., 2003;Moore and Rojstaczer, 2002;Niu et al.,
453 2007).


Many studies have reported vegetation related parameters as the second influential predictor of
groundwater recharge. Vegetation has a high correlation with other physical variables such as
soil moisture, runoff capacity and porosity, which adds to its recharge explanatory power (Kim
and Jackson, 2012;Scanlon et al., 2005). In this study recharge rate was high, where runoff
water have more retention time on the surface. This was mainly observed for shallow rooted
vegetation like grasslands. In deep rooted forest areas recharge was reduced because of
increased evapotranspiration (Kim and Jackson, 2012). However, not all reported studies are
in agreement with vegetation as an important predictor of recharge. For example, Tögl (2010)
failed to find a correlation between vegetation/land cover and recharge. This may be the result
of some peculiarity in the study dataset. Apart from the predictors discussed above, depth to
groundwater and surface drainage density were also identified as potential predictors of
recharge from literature (Döll and Flörke, 2005;Jankiewicz et al., 2005). Despite this they were
excluded from this study because of the lack of appropriate resolution global datasets.

The total recharge estimated in this study is strongly consistent with results from complex
global hydrological models. Long term average annual recharge was found to be 134 mm/yr.
The total recharge estimated in this study (13,600 km$^3$/yr) was very close to existing estimates
of complex hydrological models except those using MATSIRO, which overestimates recharge
in humid regions (Koirala et al., 2012). The results shown in Table 4 indicate that, compared
to existing techniques, the model developed in this study can make recharge assessments with
the same reliability but with fewer computational requirements. Moreover, the error in recharge
prediction in this study was low, ranging from only -8 mm/yr to 10 mm/yr for 97.2% of cases.

Table 4. Global estimates of groundwater recharge

| Model Used | Spatial Resolution | Temporal Range | Total Global Recharge ( km$^3$/yr) | Reference |
|---|---|---|---|---|
| Empirical model | 0.5deg | 1981-2014 | 13,600 | Current study |
| WaterGAP 2 | 0.5deg | 1961-1990 | 14,000 | (Döll, 2002) |
| WaterGAP | 0.5deg | 1961-1990 | 12,666 | (Döll and Flörke, 2005) |
| PCR GlobWB | 0.5deg | 1958-2001 | 15,200 | (Wada et al., 2010) |
| PCR GlobWB | 0.5deg | 1960-2010 | 17,000 | (Wada et al., 2012) |
| MATSIRO | 1deg | 1985-1999 | 29,900 | (Koirala et al., 2012) |
| FAO Statistics | Country | 1982-2014 | 10,613 | (FAO, 2016) |


The global recharge map developed showed a similar pattern to recharge maps produced using
complex global hydrological models. The results of this study indicate that recharge across the
globe was varied considerably as a function of spatial region, and was analogous to global



distribution of climate zones (Scanlon et al., 2002). Humid regions had very high recharge
compared to arid (semi-arid) regions, which is obviously due to the higher availability of water
for recharge. Recharge was also affected by climate variability and climate extremes at a
regional level (Scanlon et al., 2006;Wada et al., 2012). However, an effect of climate variability
on inter annual recharge at a global-scale was not pronounced in our results. The potential
reason for this is that the El Nino Southern Oscillation (ENSO), the primary factor that
determines climate variability globally, has converse effects in different parts of the world. The
effects of increased precipitation in some parts of the world would have been counteracted by
reductions in precipitation in other areas resulting in relatively small effect on inter annual
variation in global recharge.

## 493  5  Conclusion

This study presents a new method for identifying the major factors influencing groundwater
recharge and using them to model large scale groundwater recharge. The model was developed
using a dataset compiled from the literature and containing groundwater recharge data from
715 sites. In contrast to conventional water balance recharge estimation, a multimodel analysis
technique was used to build the model. The model developed in this study is purely empirical
and has fewer computational requirements than existing large scale recharge modelling
methods. The $0.5^0$ global recharge estimates presented here are unique and more reliable
because of the extensive validation done at different scales. Moreover, inclusion of a range of
meteorological, topographical, lithological and vegetation factors adds to the predictive power
of the model. The results of this investigation show that meteorological and vegetation factors
had the most predictive power for recharge. The high dependency of recharge on
meteorological predictors make it more vulnerable to climate change. Apart from being a
computationally efficient modelling method, the approach used in this study has some
limitations. Firstly it does not include direct anthropogenic effects on the groundwater system
and also excludes focused recharge by natural or artificial means, suggesting scope for further
future development. Secondly, the recharge data set used in this study did not include data
points from frozen regions. Therefore, Greenland and Antarctica were excluded from the final
recharge map. However, the model developed in this study and the recharge maps produced
will aid policy makers in predicting future scenarios with respect to global groundwater
availability.

## 514  6  Acknowledgement

This project was partially supported by the Australian Research Council through project
(FT130100274). The authors would like to acknowledge the University of Melbourne for
providing computational and other technical facilities for this research, and also the
international agencies that provided the data required for this study.

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
