# Peer review of "Predicting groundwater recharge for varying landcover and climate conditions: – a global meta-study"

_Hydrology and Earth System Sciences, 2017_

## Referee Comment (RC1) · Anonymous Referee #1 · 17 Jan 2018

The paper determines the most important factors controlling groundwater recharge rate, and uses a statistical model to estimate groundwater recharge globally. The work includes the creation of a global database of recharge estimates from 715 sites reported in past studies. Using a multimodel approach, the compiled database is used to create an empirical model to predict groundwater recharge at 0.5 degree resolution. The work is of interest and suitable for this journal as it deals with a relevant topic, and has the potential to contribute to future work involving large-scale hydrological phenomena by providing reasonable estimates of groundwater recharge using fewer computational resources. Overall, the writing quality of the paper is good. The transparency of the database compiled is particularly noteworthy. The paper, however, would benefit

from some revisions. In particular, the results section was at times difficult to follow. Also, some of the figures are unclear and should be modified before publication. For this reason, I suggest the editor consider the revisions suggested below prior to making a decision on this manuscript.

Specific Comments:

Line 78: The fact that the FAO estimates are limited/unreliable is mentioned twice in the paper. How so? It would useful to delve deeper into the limitations of the FAO methodology to help the readers. Lines 123-130 highlights the rationale for selecting the explanatory factors in this study. Were any relevant factors excluded due to data/other constraints? Line 341-343: What was the Vopt for the top 10 models? Are the predictors shown in Table 3 equivalent to Vopt? Vopt could also be labelled on Figure 5 to make it clear. Line 366: How did the models with R2 = 0.56 differ from the top 3 models shown in figure 5 which have a R2 of ∼0.42? Line 371-372: It might be useful to add these tests to a supplemental document Figures 3, 6 and 7 are not very clear. Increasing the size of axis text/legend would help. Figure 7 appears stretched. Line 385-413: The procedure to calculate the recharge values shown in Figures 8-11 is not very clear. Was one of the 'better' models used to calculate the map? Or, were all the 'better' models used and then averaged? Please clarify. It would also be useful to have a table that has the regression coefficients for selected models that includes the R2 values. Figure 11 compares the model estimated mean annual groundwater recharge for different countries with the FAO estimates. It would be pertinent to see if the countries that are most deviant from the 1:1 line are ones that didn't have study sites (out of the 715) used in the analysis. Line 412 and Line 480: Given that the FAO method is unreliable, how does the country-wide model results compare with estimates from complex hydrological models like PCR-GlobWB and WaterGAP? This is fairly important as it would help solidify the results obtained in the study. Line 455-467: While this paragraph discusses the influence of vegetation on recharge, the results fail to illustrate this influence. Please clarify how this influence was observed in the results.

Technical Corrections: Figure 5 is a multi-part figure and should be labelled a),b), c) The legends in Figures 8 and 10 are difficult to see Line 643-645: Citation format not consistent

---

## Referee Comment (RC2) · Anonymous Referee #2 · 9 Feb 2018

Review for HESS
4[th] February 2018

Predicting groundwater recharge for varying landcover and 2 climate conditions: – a global meta-study.

Mohan,C , Western,A.W. Wei,Y. and Saft, M.

**Summary**

This work aims to improve our understanding of the key factors which are important in estimating diffuse groundwater recharge at a global scale. An empirical model was developed using predictive variables, which characterized the meteorological, topographical, lithological and vegetation domains, all of which were determined from globally available datasets.  Results are validated against findings from 715 studies worldwide which were found in the literature, as well as statistics provided by the Food and Agriculture Organisation (FAO). In order to minimise the effects of model structure uncertainty on results, the performance of multiple models was assessed.  This work is of interest to the journal as it contributes to the understanding of large scale hydrological processes, whilst also trying to use methods that gain knowledge from globally available datasets. I believe this paper should be accepted with minor revisions. Prior to acceptance I would particularly expect the main suggestions to be addressed.

**Main Suggestions**

1. Model validation

   a) I believe this work validates modelling results against the estimations found in the literature and those of the FAO simultaneously. It could be a good exercise to validate the model using each dataset independently, to see how the use of the newly compiled information improves (hopefully) model estimations.

   b) Line 425: Figure 11 compares modelled recharge estimates to those of the FAO, why is this done, if in line 412 it states the comparison is unreliable?  Would it not be better to compare modelled results to those of the 715 recharge sites? I think this is particularly important, as the compilation of this information to validate model results is one of the key things which separates this work out from others. Hence, I would like to see how they compare to each other.

2. Selection of model predictors

   a) Lines 123-130: Would benefit from explaining the rationale used in selecting the potential predictors further (especially as it is deemed a "key step"), i.e. why is the number of rainy days important? Why were mean precipitation and potential evapotranspiration selected as well as aridity index?

**Minor Suggestions**

1.  Lines 76-79: Questions the reliability of the FAO estimates. Please make it clear why these estimates are unreliable. How are they derived?

2.  Line 79: States no study has previously validated modelled estimates against small scale recharge estimates. However, Doll and Fiedler (2008) used local recharge estimates to test the performance and modify the algorithm used to determine recharge for arid and semi-arid cells.

3.  Line 109: Would be interesting to know how the use of different recharge estimation methods found in the literature varied spatially and why. Could be shown graphically.

4.  Line 118: Were certain climates or land uses over or under represented by the 715 recharge estimation sites? Is there an inherent bias in the dataset collected? A histogram could be useful.

5.  Line 114: Recharge estimates in the literature may be representative of different time periods, especially if they were determined via water balance are water table fluctuation methods. However, the model predictors and the modelled recharge estimates are given as a mean for the period of 1981 and 2014. How was the inconsistency in the timeframe of the data managed? How did it effect model validation using the new dataset?

6.  Lines 127-128: Were there any predictors which you would have liked to use, but were not available from the global datasets?

7.  Line 201: I'm uncertain whether there were predictors which were rejected prior to the main bulk of the work. i.e. were there initially more predictors than shown in Table 1, with those in Table 1 just being those accepted for use?

8.  Line 284: States that maps illustrating the percentage of rainfall becoming recharge were generated. However, these are not shown in this work.

9.  Line 287: Refers to the koopan classification which I believe is meant to be $K\ddot{o}ppen - Geiger$.

10. Line 415: Section 2.3 states that Figure 8 (global recharge estimation map) was derived from the best model found. It would be good to repeat this in the Figure heading "Best model estimation".

11. Line 415: Interesting to see some of the regions where greater recharge estimates are determined (South America, Indonesia) also coincide with areas which are less represented by the 715 studies. How uncertain are results in these areas? Could the uncertainty of these estimates be assessed?

12. Line 417: Figure 9 clearly indicates the importance of mean annual precipitation for mean annual diffuse recharge at the global scale. It would be interesting to contrast this to the relationship between mean annual precipitation and the annual recharge rates reported in the studies, in order to illustrate whether the influence of meteorology on groundwater recharge is site specific.

13. Line 486: Is this work able to say whether there are regions in the world which have declining or augmenting rates of recharge in the 1981-2014 time period?

14. Figures and Tables

    a) Generally, the figure style should be more consistent. Some figures don't have extents marked out (1, 2, 6, 7, 8, 10) whereas others do (3, 4, 5, 9, 11).
    b) Figure sizes should be more consistent; Figure 3 appears smaller than Figure 4, Figure 6 appears smaller than Figure 7 and Figure 8 appears smaller than Figure 9.
    c) Figures sitting one above the other should sit more squarely to one another (6 and 7, 8 and 9).
    d) Figure 5 is made up of multiple figures and therefore would benefit from a, b, c notation.
    e) Axis labels, legends and titles above map figures are often too small to read.

15. References

    a) Please include website addresses in reference of websites (Line 574: AQUASTAT)

16. Readability

    a) Generally, this paper is well written, making it easy to read and understand key concepts.

**References**

Doll,P. and Fiedler,K. (2008). Global-scale modeling of groundwater recharge. Hydrol. Earth Syst. Sci., 12, 863–885

---

## Author Comment (AC1) · 11 Feb 2018

**Response to Anonymous Referee #1's comments on manuscript hess-2017-679 (Predicting groundwater recharge for varying landcover and climate conditions: – a global meta-study)**

We sincerely thank Anonymous Referee #1 for his/her constructive comments which have helped to improve the article. We address each comment in turn below.

Comment: Line 78: ***The fact that the FAO estimates are limited/unreliable is mentioned twice in the paper. How so? It would useful to delve deeper into the limitations of the FAO methodology to help the readers.***

We have added the following to Lines 78-84 to clarify this.

FAO statistics were based on estimates compiled from national institutions. The data estimation and reporting capacities of national agencies vary significantly and raise concerns about the accuracy of the data (Kohli and Frenken, 2015). In addition, according to FAO AQUASTAT reports, most national institutions in developing countries prioritise subnational level statistics over national level statistics, and in most cases data is not available for all sub national entities. This decreases the accuracy of country wide averages and raises concerns about the reliability of using them as standard comparison measures.

Comment: Lines 123-130: ***highlights the rationale for selecting the explanatory factors in this study. Were any relevant factors excluded due to data/other constraints?***

It is true that insufficient and poor quality data often limit studies such as ours, and we have amended the relevant paragraph to acknowledge this more clearly. The relevant section of the paragraph now reads:

Line 134 -142: The choice of predictors was made based on the availability of global gridded datasets and their relative importance in a physical sense, as informed by the literature. We employed 12 predictors comprising meteorological factors, soil/vadose zone factors, vegetation factors and topographic factors. However, other factors which could have a sizable influence on recharge were not included in this study because there was insufficient data. Given this, we did not consider the effects of irrigation on recharge, limiting the scope of the study to rainfall induced recharge. Subsurface lithology can be another major factor determining recharge. Once again, due to the lack of lithological and geological datasets at a larger scale, these factors were also eliminated from the study. Better quality information about various predictors would have been desirable to enhance the accuracy of prediction.

Comment: Line 341-343: ***What was the Vopt for the top 10 models? Are the predictors shown in Table 3 equivalent to Vopt? Vopt could also be labelled on Figure 5 to make it clear.***

We have made some changes in terminology improve the clarity of this aspect of the paper. Figure 5 is changed as a result and the discussion is modified as follows.

Line 352-359: The choice of better models was made by considering the PoE of individual predictors (refer section 3.2.1) and the number of predictors in the model ($V$). Figure 5 shows the performance criteria for the top three models for different $V$ values. The model performance increased with $V$ up to 4 to 6 depending on the different criteria. After that, AICc, CAIC, RMSE and $R^2_{adj}$ values remained almost constant, indicating that further addition of predictors did not improve the model performance. In particular CAIC shows a clear minimum at V=4 and it penalises model complexity

more rigorously. Table 3 illustrates the predictors in the top 10 models selected based on CAIC. Nine of the top 10 models had $V=4$, and the remaining model (3rd best) had $V=3$.

[Figure]

Figure 5. (a)CAIC (b) RMSE, and (c) $R^2$adj for the top 3 models with different number of predictors upto 12 and the green dotted lines representing the number of predictors for the best performance criteria value.

Comment: Line 366: ***How did the models with $R^2$ = 0.56 differ from the top 3 models shown in figure 5 which have a $R^2$ of ~0.42?***

The top three models shown in Figure 5 were built using the entire dataset, whereas the models discussed in line 366 (with $R^2$=0.56) were built as part of sub sample testing. In the sub sample testing, the entire dataset was randomly divided into 80% and 20% subsets and 80% of the data were used for building the model. The predictive performance developed model was tested against the omitted 20% of data (Reff line 233 -236). The statistics mentioned in Line 366 correspond to only 20% of the data at a time.

Comment: ***Figures 3, 6 and 7 are not very clear Increasing the size of axis text/legend would help. Figure 7 appears stretched.***

We have amended the figures as suggested, as shown below:

[Figure]

Figure 3. Model fit performance criteria for single predictor regressions.

[Figure]

Figure 6. RMSE of sub-sample (a) model fitting and (b) model prediction of top 10 models according to CAIC.

[Figure]

Figure 7 (a) Error at each data point along with the corresponding rainfall obtained using the leave-one-out model testing procedure and (b) Scatter plot between error at each data point and corresponding precipitation.

Comment: Line 385-413: ***The procedure to calculate the recharge values shown in Figures 8-11 is not very clear. Was one of the 'better' models used to calculate the map? Or, were all the 'better' models used and then averaged? Please clarify. It would also be useful to have a table that has the regression coefficients for selected models that includes the R2 values.***

We have tried to clarify the method and the relevant text (lines 398-401) now reads as follows:

In this study, the best model as defined by CAIC (model 1 in Table 3) was used to generate the recharge map. However, due to the similarity in structure of the top 10 models (Table 3), all models were equally good at predicting groundwater recharge and gave similar results (not shown).

We have revised Table 3 by adding model parameter coefficients and Adj $R^2$ values as shown below:

Table 3. Coefficient of predictors used in the top 10 models, ranked based on CAIC.

| Model Rank | P | T | PET | Rd | S | $k_{sat}$ | SWSC | AI | EW | $\rho_b$ | Clay | LU | $R^2_{adj}$ |
|---|---|---|---|---|---|---|---|---|---|---|---|---|---|
| 1 | 0.008 | | -0.005 | | | 0.008 | | | | | | -0.915 | 0.407 |
| 2 | 0.009 | | -0.005 | | | | | | | | -0.062 | -0.887 | 0.407 |
| 3 | 0.008 | | 0.005 | | | | | | | | | -0.862 | 0.401 |
| 4 | 0.007 | | -0.004 | | | | | 0.002 | | | | -0.050 | 0.253 |
| 5 | 0.008 | | 0.020 | | | | | | | 0.250 | | -0.003 | 0.332 |
| 6 | 0.009 | | -0.005 | | | | | | -0.062 | | | -0.887 | 0.410 |
| 7 | 0.009 | 0.003 | -0.171 | | | | | | | | | -2.258 | 0.407 |
| 8 | 0.007 | | -0.004 | | | | | 0.002 | | | | -2.014 | 0.346 |
| 9 | 0.008 | | -0.005 | -0.032 | | | | | | | | -0.864 | 0.335 |
| 10 | 0.008 | | -0.005 | | -0.001 | | | | | | | -0.052 | 0.404 |

Comment: *Figure 11 compares the model estimated mean annual groundwater recharge for different countries with the FAO estimates. It would be pertinent to see if the countries that are most deviant from the 1:1 line are ones that didn't have study sites (out of the 715) used in the analysis.*

We have added a new figure (Figure 12) and expanded the discussion accordingly.

Figure 12 shows the country wide distribution of errors in model prediction in comparison with FAO statistics. Very high errors were found in countries with fewer model building data points. The model considerably overestimated recharge for Russia, Canada, Brazil, Indonesian Malaysia and Madagascar.

[Figure]

**Groundwater Recharge Residual**

**Legend**

♦ RechargeDataPoints

**Residual (mm/yr)**

| | |
|---|---|
| ■ (dark red) | -1020.008019 - -172.880000 |
| ■ (red) | -172.879999 - -98.129000 |
| ■ (dark orange) | -98.128999 - -68.347000 |
| ■ (orange) | -68.346999 - -42.650000 |
| ■ (pale yellow-green) | -42.649999 - -16.210000 |
| ■ (light green) | -16.209999 - -4.137000 |
| ■ (green) | -4.136999 - 4.857000 |
| ■ (teal) | 4.857001 - 10.000000 |
| ■ (cyan) | 10.000001 - 50.540000 |
| ■ (blue) | 50.540001 - 4877.870202 |

Figure 12. Spatial distribution of groundwater recharge residuals (FAO estimates – Model estimates) along with recharge sites selected for model building.

Comment: Line 412 and Line 480: *Given that the FAO method is unreliable, how does the country-wide model results compare with estimates from complex hydrological models like PCR-GlobWB and WaterGAP? This is fairly important as it would help solidify the results obtained in the study*

We have added a new figure (Figure 11 (b)) comparing country level recharge estimates from the current model with WaterGAP, and revised the discussion accordingly. We were not able to compare our results with PCR-GlobWB, as its country-wide recharge results are not publically available.

We added the following to Line 428-431: Recharge estimates from the best models in the present study were compared to recharge estimates from the complex hydrological model (WaterGAP). Even though the model in this study overestimates recharge for countries with fewer data points, the scatter shows a smaller spread compared to the FAO estimates.

[Figure]

Figure 11. Comparison of predicted recharge against country level estimates from (a) FAO and (b) WaterGAP model

Comment: Line 455-467: *While this paragraph discusses the influence of vegetation on recharge, the results fail to illustrate this influence. Please clarify how this influence was observed in the results.*

We have modified the discussion by highlighting the importance of vegetation as shown in our results.

Line 493 -497: In this study Land Use (LU) was used as a proxy for vegetation. According to the results, LU was found to be one of the predictors having the highest Proportion of Evidence (PoE) (Figure 4). In addition, all the better performing models included LU as one of the predictors which clearly indicates that vegetation is one of the most influential factors for groundwater recharge.

*Technical Comments*

*Figure 5 is a multi-part figure and should be labelled a),b), c) The legends in Figures 8 and 10 are difficult to see Line 643-645: Citation format not consistent*

We have improved Figures, 5, 8 and 10 as shown below:

[Figure]

Modified Figure 8. Long-term (1981 -2014) average annual groundwater recharge estimated using the developed model.

[Figure]

[Figure]

Figure 10. Map showing (a) coefficient of variability and (b) standard deviation of annual groundwater recharge from 1981 to 2014.

We have revised the citation format as indicated below:

Kohli, A. and Frenken, K.: Renewable Water Resources Assessment – 2015 AQUASTAT methodology review, Food and Agricultural Organisation of the United Nations, 1-6 pp., 2015.

---

## Author Comment (AC2) · 25 Feb 2018

**Response to Anonymous Referee #2's comments on manuscript hess-2017-679 (Predicting groundwater recharge for varying landcover and climate conditions: – a global meta-study)**

We sincerely thank Anonymous Referee #2 for their constructive comments, which have helped to improve the article. We address each comment in turn below.

**Comment: a) I believe this work validates modelling results against the estimations found in the literature and those of the FAO simultaneously. It could be a good exercise to validate the model using each dataset independently, to see how the use of the newly compiled information improves (hopefully) model estimations.**

We tested model predictions in three ways using two data sets: estimates from the recharge studies we collated; and estimates from FAO. First, we undertook a cross-validation of the model structure and predictor selection by setting aside 20 percent of the recharge studies and repeating that exercise. Second, we undertook a leave-one-out cross validation of the individual predictions, also using the collated recharge studies. Both these are independent because the validation points were not used in model fitting, as is standard practice for cross-validation studies. This is explained in the methodology. Third, we compared our model predictions against FAO country level statistics. The country level statistics were not used at any other point in the study. Finally, in response to Referee #1, we also plan to add a comparison against a global model. All of this represents a significant effort in independent validation, most of which is documented in the original paper. The validations described above were actually implemented on each data set independently in the original paper. We will review the wording of the methodology to ensure that the independent nature of this validation is conveyed more clearly.

**Comment: b) Line 425: Figure 11 compares modelled recharge estimates to those of the FAO, why is this done, if in line 412 it states the comparison is unreliable? Would it not be better to compare modelled results to those of the 715 recharge sites? I think this is particularly important, as the compilation of this information to validate model results is one of the key things which separates this work out from others. Hence, I would like to see how they compare to each other.**

We agree that comparing the model results against FAO is not an ideal validation of the results, although it is a useful comparison to understand the differences between the data sets. The FAO data were used for a country level validation because there are no other non-modelled large scale estimates, as far as we are aware. In section 3.2.3, the model results were compared against the literature compiled groundwater recharge. We did actually compare results against the 715 recharge sites via a leave-one-out cross validation in Figure 7.

In response to Referee 1's comments, we also undertook a comparison against global model estimates of recharge and will add a figure comparing the modelled recharge against the recharge estimates from a global hydrological model (WaterGAP), together with the following text

Line 428-431:

Recharge estimates from the best models in the present study were compared to recharge estimates from the complex hydrological model (WaterGAP). Even though the model in this study overestimates recharge for countries with fewer data points, the scatter shows a smaller spread compared to the FAO estimates.

[Figure]

Figure 11. Comparison of predicted recharge against country level estimates from (a) FAO and (b) WaterGAP model.

**2. Selection of model predictors**
**a) Lines 123-130: Would benefit from explaining the rationale used in selecting the potential predictors further (especially as it is deemed a "key step"), i.e. why is the number of rainy days important? Why were mean precipitation and potential evapotranspiration selected as well as aridity index?**

We have amended the relevant paragraph to explain this more clearly. The relevant section of the paragraph now reads (Line 134-140):

Recharge depends on drainage from the soil profile and the partitioning of that drainage between shallow lateral flow to streams and deep drainage to the water table. A variety of factors influence this including the meteorological forcing, the properties of the soil profile, the properties of the vegetation, and the topography (Scanlon et al., 2002; Kim and Jackson, 2012;Scanlon et al., 2005). Vegetation is important particularly in its influence on partitioning of available water to evapotranspiration (Zhang et al., 2001) and we selected land use to represent this. The most important soil properties are likely to relate to the soil water storage capacity and drainage properties of the soil profile as they represent the capacity of the soil to buffer meteorological forcing variation and the capacity of the soil to transmit water to depth(Scanlon et al., 2005). We chose hydraulic

conductivity, soil water storage capacity, clay content and bulk density as surrogates for these soil hydraulic properties. Partitioning of drainage between vertical and lateral is likely to depend on both the existence or otherwise of impeding layers in the vadose zone profile and on the topographic slope (Saffarpour et al., 2016). We included slope as a predictor but could find no specific information beyond the above soil properties on drainage impediments.

Focussing on the meteorological aspects, drainage depends on there being an excess of water availability in the soil profile and hence on variations in precipitation and evapotranspiration at both short-term (hours and days), seasonal timescales and longer. This suggests that precipitation, evaporative forcing and aridity index are obvious candidate predictors, although we should only expect a subset to prove valuable in the final model as there is shared information between them. In addition, the concentration of the precipitation in time and the relationship between precipitation of evapotranspiration seasonally also influence the occurrence of drainage. The number of rain days was considered as it potentially provides useful information on how concentrated the rainfall is in time. The seasonality of soil water content and drainage is also strong (Grayson et al., 1997;Western et al., 2004) and is typically driven by a seasonal excess of precipitation over potential evapotranspiration, which we aim to capture using the Excess Water predictor.

A set of 12 predictors comprising meteorological factors, soil/vadose zone factors, vegetation factors and topographic factors was finally selected (Table 1). Data for these corresponding to 715 recharge study sites were extracted from…..

**Minor Suggestions**
**1. Lines 76-79: Questions the reliability of the FAO estimates. Please make it clear why these estimates are unreliable. How are they derived?**

We have added the following to Lines 78-84 to clarify this.

FAO statistics are based on estimates compiled from national institutions. The data acquisition and reporting capacities of national agencies varies significantly and this raises concerns about the accuracy of the data (Kohli and Frenken, 2015). In addition, according to the FAO AQUASTAT reports, most national institutions in developing countries prioritise subnational level statistics over national level statistics, and in most cases data is not available for all sub national entities. This decreases the accuracy of country wide averages and raises concerns about the reliability of using them as standard comparison measures.

**2. Line 79: States no study has previously validated modelled estimates against small scale recharge estimates. However, Doll and Fiedler (2008) used local recharge estimates to test the performance and modify the algorithm used to determine recharge for arid and semi-arid cells.**

We agree that Doll and Fiedler (2008) used 51 observed recharge estimates for tuning the model results and the comparison is restricted to arid and semi-arid zones only. We have modified our text at line 79 as follows:

Line 84-88: Very few large-scale studies had validated modelled estimates against small scale recharge measurements. Döll and Fiedler (2007) used 51 recharge observations from arid/semi-arid regions for adjusting the model outputs. In comparison to it, this study does a more extensive validation of the model against 715 local recharge measurements.

**3. Line 109: Would be interesting to know how the use of different recharge estimation methods found in the literature varied spatially and why. Could be shown graphically.**

We modified Figure 1 to represent recharge estimation methods spatially and added the following text (Line 112-115):

The final recharge data set consisted of 715 data points spread across the globe (Figure 1). Of these studies, 345 were estimated using the tracer method, 123 using the water balance method, and the remaining studies used base flow, lysimeter, or water table fluctuation methods

[Figure]

**Legend**

**Recharge Estimation Methods**

- Water table fluctuation method
- Tracer method
- Water balance method
- Lysimeter
- Model
- Baseflow separation method

Figure 1. Locations of the 715 selected recharge estimation sites used for model building, together with the corresponding recharge estimation method.

**4. Line 118: Were certain climates or land uses over or under represented by the 715 recharge estimation sites? Is there an inherent bias in the dataset collected? A histogram could be useful.**

We have modified Figure 2 and expanded the discussion accordingly.

[Figure]

Figure 2. Histogram showing (a) frequency and spread of year of study and distribution of recharge estimates across different (b) Land Use and different (c) Climate zones based on Koppen Geiger climate classification.

Line 122-126: Moreover, the compiled dataset does not represent all climate zones well (Figure 2 (c)), with most studies being in either arid, semi-arid or temperate zones. Pasture and cropland were the dominating land uses in the dataset (Figure 2(b)).

**5. Line 114: Recharge estimates in the literature may be representative of different time periods, especially if they were determined via water balance are water table fluctuation methods. However, the model predictors and the modelled recharge estimates are given as a mean for the period of 1981 and 2014. How was the inconsistency in the timeframe of the data managed? How did it effect model validation using the new dataset?**

One of the major limitation of this study is the inconsistency in time frame of the estimates (line 111 -114). Overall the recharge measurements spaned 34 years from 1981 to 2014 and study lengths varied from 1981 to 2014 years. When it comes to globally scattered measurements, it was practically impossible to get consistent data spatially and temporally. Therefore, in this study, both for model building and validation, variables were averaged over the period of 34 years to minimize the inconsistency. The averaging of the variables can introduce bias in the prediction, especially in extreme recharge areas. The higher recharge may be slightly under predicted and the lower recharge may be slightly over predicted.

**6. Lines 127-128: Were there any predictors which you would have liked to use, but were not available from the global datasets?**

It is true that insufficient and poor quality data often limit studies such as ours, and we have amended the relevant paragraph to acknowledge this more clearly. The relevant section of the paragraph now reads:

Line 134 -142: The choice of predictors was made based on the availability of global gridded datasets and their relative importance in a physical sense, as informed by the literature. We employed 12

predictors comprising meteorological factors, soil/vadose zone factors, vegetation factors and topographic factors. However, other factors which could have a sizable influence on recharge were not included in this study because there was insufficient data. These included: the effects of irrigation on recharge, limiting the scope of the study to rainfall induced recharge; and subsurface lithology, which may be another important factor determining recharge.

**7. Line 201: I'm uncertain whether there were predictors which were rejected prior to the main bulk of the work. i.e. were there initially more predictors than shown in Table 1, with those in Table 1 just being those accepted for use?**

As mentioned in the above response, some of the predictors were eliminated from the study due to data unavailability. Particularly all irrigation factors and geology factors were excluded because of lack of proper datasets. Other than that, we did not eliminate any predictors prior to the main bulk of work.

**8. Line 284: States that maps illustrating the percentage of rainfall becoming recharge were generated. However, these are not shown in this work.**

The maps were not included in the final manuscript to reduce the length of the final draft. In response to this comment, we decided to include the following figure in the supplementary material.

**9. Line 287: Refers to the koopan classification which I believe is meant to be Koppen Geiger.**

Yes we meant Köppen-Geiger classification. The following changes were made to reflect this:

Line 314-315: As recharge data from regions with frozen soil were scarce in the model building dataset, the model predictions in those regions particularly for regions with Köppen-Geiger classification Dfc, Dfd, ET and EF are not highly reliable, so the EF regions of Greenland and Antarctica were excluded due to lack of data.

**10. Line 415: Section 2.3 states that Figure 8 (global recharge estimation map) was derived from the best model found. It would be good to repeat this in the Figure heading "Best model estimation".**

Figure 8 is changed as given below

[Figure]

Figure 8. Long-term (1981 -2014) average annual groundwater recharge estimated using the best model.

**11. Line 415: Interesting to see some of the regions where greater recharge estimates are determined (South America, Indonesia) also coincide with areas which are less represented by the 715 studies. How uncertain are results in these areas? Could the uncertainty of these estimates be assessed?**

We acknowledge this is an issue and have addressed it by adding the following:

Line 531-535: Uncertainty in recharge estimates is likely to increase in areas with poor data coverage, which tend to be those that are wetter, both in the tropics and cold regions. While we would expect this to be the case, it is surprising that the residual analysis from the cross validation (Figure 7b) suggests uncertainty does not grow particularly rapidly with precipitation, at least up to 1500mm/year.

**12. Line 417: Figure 9 clearly indicates the importance of mean annual precipitation for mean annual diffuse recharge at the global scale. It would be interesting to contrast this to the relationship between mean annual precipitation and the annual recharge rates reported in the studies, in order to illustrate whether the influence of meteorology on groundwater recharge is site specific.**

Please refer to the following lines in the manuscript which answers the above comment.

line 498 - 506: In most cases, especially dry regions, groundwater recharge is controlled by the availability of water at the surface, which is mainly controlled by precipitation, evapotranspiration and geomorphic features (Scanlon et al., 2002). Numerous studies agree with this finding. For example, in south western USA, 80% of observed recharge variation is explained by mean annual precipitation (Keese et al., 2005). However, the influence of meteorological factors on groundwater recharge is highly site-specific (Döll and Flörke, 2005). The effect of meteorological factors can also depend on whether the season or year is wet or dry, type of aquifer and irrigation intensity (Adegoke et al., 2003;Moore and Rojstaczer, 2002;Niu et al., 2007).

**13. Line 486: Is this work able to say whether there are regions in the world which have declining or augmenting rates of recharge in the 1981-2014 time period?**

To address this comment, the following figures showing inter decadal percentage change in groundwater recharge have been added in the supplementary material.

It is possible to say using the model whether the regions have declining or augmenting recharge rates. Hence the model is highly influenced by the changes in precipitation, the inter annual changes in the recharge will be highly correlated to that in precipitation. The following figures are added in the supplementary material and the following paragraph is added to explain this idea further.

Line 434 – 445: It is also interesting to consider trends in recharge over time. Our model includes both meteorology and landuse as predictors that can change in time and so can produce estimates of change in response to these variables only. We estimated recharge on a decade-by-decade basis based on meteorological fluctuations only and then calculated percentage change between the decades (Figure S2 Supplementary). These maps show some distinct regional patterns that appear to reflect a) linear features where there are strong and shifting regional gradients such as the African Sahel (Giannini et al., 2008) that show as distinct linear features and b) more general regional to continental scale changes, such as in Australia, which was strongly affected by the Millennium Drought

(van Dijk et al., 2013) and associated climate fluctuations.  These results suggest that inter-decadal variability in groundwater recharge may be quite large in many regions.

**Percentage Change in Recharge (first decade to second decade)**

[Figure]

**Percentage Change in Recharge (second decade to third decade)**

[Figure]

Figure S2. Map showing change in mean percent decadal recharge (a) from 1981-1990 to 1991-2000 and (b) from 1991-2000 to 2001-2010. (Decadal change = mean decadal recharge of later decade – mean decadal recharge of former decade).

Reference

Adegoke, J. O., Pielke Sr, R. A., Eastman, J., Mahmood, R., and Hubbard, K. G.: Impact of irrigation on midsummer surface fluxes and temperature under dry synoptic conditions: A regional atmospheric model study of the US High Plains, Monthly Weather Review, 131, 556-564, 2003.
Döll, P., and Flörke, M.: Global-Scale estimation of diffuse groundwater recharge: model tuning to local data for semi-arid and arid regions and assessment of climate change impact, 2005.
Döll, P., and Fiedler, K.: Global-scale modeling of groundwater recharge, Hydrology and Earth System Sciences Discussions, 4, 4069-4124, 2007.

Giannini, A., Biasutti, M., and Verstraete, M. M.: A climate model-based review of drought in the Sahel: Desertification, the re-greening and climate change, Glob. Plan. Change, 64, 119-128, 10.1016/j.gloplacha.2008.05.004, 2008.

Grayson, R. B., Western, A. W., Chiew, F. H. S., and Blöschl, G.: Preferred states in spatial soil moisture patterns: Local and non-local controls, Water Resour. Res., 33, 2897-2908, 1997.

Keese, K., Scanlon, B., and Reedy, R.: Assessing controls on diffuse groundwater recharge using unsaturated flow modeling, Water Resources Research, 41, 2005.

Kohli, A., and Frenken, K.: Renewable Water Resources Assessment – 2015 AQUASTAT methodology review, Food and Agricultural Organisation of the United Nations, 1-6, 2015.

Moore, N., and Rojstaczer, S.: Irrigation's influence on precipitation: Texas High Plains, USA, Geophysical Research Letters, 29, 2002.

Niu, G. Y., Yang, Z. L., Dickinson, R. E., Gulden, L. E., and Su, H.: Development of a simple groundwater model for use in climate models and evaluation with Gravity Recovery and Climate Experiment data, Journal of Geophysical Research: Atmospheres, 112, 2007.

Saffarpour, S., Western, A. W., Adams, R., and McDonnell, J. J.: Multiple runoff processes and multiple thresholds control agricultural runoff generation, Hydrol. Earth System Sci., 20, 4525-4545, 10.5194/hess-20-4525-2016, 2016.

Scanlon, B. R., Healy, R. W., and Cook, P. G.: Choosing appropriate techniques for quantifying groundwater recharge, Hydrogeology journal, 10, 18-39, 2002.

van Dijk, A. I. J. M., Beck, H. E., Crosbie, R. S., de Jeu, R. A. M., Liu, Y. Y., Podger, G. M., Timbal, B., and Viney, N. R.: The Millennium Drought in southeast Australia (2001–2009): Natural and human causes and implications for water resources, ecosystems, economy, and society, Water Resour. Res., 49, 1040-1057, 10.1002/wrcr.20123, 2013.

Western, A. W., Zhou, S.-L., Grayson, R. B., McMahon, T. A., Blöschl, G., and Wilson, D. J.: Spatial correlation of soil moisture in small catchments and its relationship to dominant spatial hydrological processes, J. Hydrol., 286, 113-134, 2004.

Zhang, L., Dawes, W. R., and Walker, G. R.: Response of mean annual evapotranspiration to vegetation changes at catchment scale, Water Resour. Res., 37, 701-708, 2001.

---

## Author Comment (AC3) · 27 Feb 2018

We thank you for your suggestions for improving the figures. All figures are modified as per the suggestions and are included in the revised manuscript.

---

## Author Response (AR1)

**Point to point response to the comments on manuscript hess-2017-679 (Predicting groundwater recharge for varying landcover and climate conditions: – a global meta-study)**

In this document, the reviewers' comments (bold font) are followed by the changes made in the final manuscript (normal font). The line numbers given in the responses are according to marked-up manuscript version.

**Line 78: The fact that the FAO estimates are limited/unreliable is mentioned twice in the paper. How so? It would useful to delve deeper into the limitations of the FAO methodology to help the readers.**
We have added the following to Lines 78-84 to clarify this.
FAO statistics were based on estimates compiled from national institutions. The data estimation and reporting capacities of national agencies vary significantly and raise concerns about the accuracy of the data (Kohli and Frenken, 2015). In addition, according to FAO AQUASTAT reports, most national institutions in developing countries prioritise subnational level statistics over national level statistics, and in most cases data is not available for all sub national entities. This decreases the accuracy of country wide averages and raises concerns about the reliability of using them as standard comparison measures.

**Line 79: States no study has previously validated modelled estimates against small scale recharge estimates. However, Doll and Fiedler (2008) used local recharge estimates to test the performance and modify the algorithm used to determine recharge for arid and semi-arid cells.**

The following lines are added to the manuscript (Line 84-88)

Only a few studies have validated modelled estimates against small scale recharge measurements. Döll and Fiedler (2007) used 51 recharge observations from arid and semi-arid regions to correct model outputs. This study develops a recharge model and undertakes a more extensive validation of it using 715 local recharge measurements.

**Line 109: Would be interesting to know how the use of different recharge estimation methods found in the literature varied spatially and why. Could be shown graphically.**

Figure 1 is modified as below in order to spatially represent different recharge estimation methods

Figure 1. Locations of the 715 selected recharge estimation sites and the corresponding recharge estimation methods, used for model building.

**Line 118: Were certain climates or land uses over or under represented by the 715 recharge estimation sites? Is there an inherent bias in the dataset collected? A histogram could be useful.**

We have modified Figure 2 and expanded the discussion accordingly.

[Figure]

Figure 2. Histograms showing frequency of (a) study year (b) Land Use and (c) Köppen–Geiger Climate zones for the recharge estimates used.

Line 124-126: Moreover, the compiled dataset does not represent all climate zones well (Figure 2 (c)), as most of the studies used were done either in arid, semi-arid or temperate zones. Pasture and cropland were the dominant land uses in the dataset (Figure 2(b)).

**Lines 123-130: highlights the rationale for selecting the explanatory factors in this study. Were any relevant factors excluded due to data/other constraints?**

**Lines 127-128: Were there any predictors which you would have liked to use, but were not available from the global datasets?**

Line 140 – 156: The choice of predictors was made based on the availability of global gridded datasets and their relative importance in a physical sense, as informed by the literature. According to the literature, the water availability on the surface for infiltration and the potential of the subsurface system to intake water are the two major controls on recharge. Different variables that can potentially represent these two factors were chosen as predictors in this study. The water availability is represented mainly by using meteorological predictors including precipitation, potential evapotranspiration, aridity index, number of days with rainfall and vegetation characteristics (land use land cover). Whereas, the intake potential is represented using various quantifiable characteristics of the vadose zone. We employed 12 predictors comprising meteorological factors, soil/vadose zone factors, vegetation factors and topographic factors. However, other factors which could have a sizable influence on recharge were not included in this study because of insufficient data. Given this, we did not consider the effects of irrigation on recharge, limiting the scope of the study to rainfall induced recharge. Subsurface lithology which could be another important recharge factor, was also eliminated from the study, due to a lack of suitable lithological and geological datasets at a larger scale. Better quality information about various predictors would have been desirable to enhance the accuracy of prediction.

**Line 341-343: What was the Vopt for the top 10 models? Are the predictors shown in Table 3 equivalent to Vopt? Vopt could also be labelled on Figure 5 to make it clear.**

We have made some changes in terminology improve the clarity of this aspect of the paper. Figure 5 is changed as a result and the discussion is modified as follows.

Line 369-378: The choice of better models was made by considering the PoE of individual predictors (refer section 3.2.1) and the number of predictors in the model ($V$). Figure 5 shows the performance criteria for the top three models for different $V$ values. The model performance increased with $V$ up to 6 to 7 depending on the different criteria. After that, AICc, CAIC, RMSE and $R^2_{adj}$ values remained almost constant, indicating that further addition of predictors did not improve the model performance. In particular CAIC reaches a minimum at V=7 and it penalises model complexity more rigorously. Table 3 illustrates the predictors in the top 10 models selected based on CAIC. All the top 10 models had $V <=7$. $P$, $PET$ and $LU$ repeatedly appeared in the predictor list of the top ten models substantiating their high predictive capacity, and the top ranked model includes these three predictors only.

[Figure]

Figure 5. (a) $R^2$adj (b) CAIC and (c) RMSE for the top 3 models with different number of predictors up to 12 and the green dotted lines representing the number of predictors for the best performance criteria value.

**The procedure to calculate the recharge values shown in Figures 8-11 is not very clear. Was one of the 'better' models used to calculate the map? Or, were all the 'better' models used and then averaged? Please clarify. It would also be useful to have a table that has the regression coefficients for selected models that includes the R2 values.**

We have tried to clarify the method and the relevant text (lines 417-420) now reads as follows: In this study, the best model as defined by CAIC (model 1 in Table 3) was used to generate the recharge map. However, due to the similarity in structure of the top 10 models (Table 3), all models were equally good at predicting groundwater recharge and gave similar results (not shown).

We have revised Table 3 by adding model parameter coefficients and Adj $R^2$ values as shown below:

Table 3. Coefficient of predictors used in the top 10 models, ranked based on CAIC.

| P | T | PET | Rd | S | $k_{sat}$ | SWSC | AI | EW | $\rho_b$ | Clay | LU | Constant | $R^2_{adj}$ |
|---|---|---|---|---|---|---|---|---|---|---|---|---|---|
| 0.0081 | | -0.0043 | | | | | | | | | 0.9567 | 5.3539 | 0.35 |
| 0.0086 | | -0.0044 | | | | | | | | -0.0606 | 1.0335 | 6.3781 | 0.25 |
| 0.0078 | | -0.0041 | | | | | | | -1.9083 | | 0.9667 | 7.8822 | 0.25 |
| 0.0076 | | -0.0055 | -0.0247 | | 0.0089 | | | 0.0040 | -2.5857 | | 1.0131 | 11.8652 | 0.34 |
| 0.0084 | | -0.0053 | -0.0195 | | | | | 0.0036 | | -0.0758 | 1.0189 | 9.4112 | 0.33 |
| 0.0092 | | -0.0052 | -0.0128 | | | | | | | -0.0631 | 1.0409 | 8.2317 | 0.33 |
| 0.0075 | | -0.0050 | -0.0194 | | | | | 0.0034 | -2.3410 | | 0.9370 | 11.2147 | 0.35 |
| 0.0084 | | -0.0049 | -0.0130 | | | | | | -2.0104 | | 0.9716 | 9.8549 | 0.35 |
| 0.0086 | | -0.0050 | -0.0122 | | | | | | | | 0.9607 | 7.0692 | 0.33 |
| 0.0086 | | -0.0053 | -0.0166 | | 0.0075 | | | | -2.1688 | | 1.0402 | 10.2082 | 0.33 |

**Figure 11 compares the model estimated mean annual groundwater recharge for different countries with the FAO estimates. It would be pertinent to see if the countries that are most deviant from the 1:1 line are ones that didn't have study sites (out of the 715) used in the analysis.**

We have added a new figure (Figure 12) and expanded the discussion accordingly.

Line 450-454: Figure 12 shows the country wide distribution of errors in model prediction in comparison with FAO statistics. Very high errors were found in countries with fewer model building data points. The model considerably overestimated recharge for Russia, Canada, Brazil, Indonesian Malaysia and Madagascar

[Figure]

[Figure]

Figure 12. Spatial distribution of groundwater recharge residual (FAO estimates – Model estimates) along with recharge sites selected for model building.

**Line 412 and Line 480: Given that the FAO method is unreliable, how does the country-wide model results compare with estimates from complex hydrological models like PCR-GlobWB and WaterGAP? This is fairly important as it would help solidify the results obtained in the study**

We have added a new figure (Figure 11 (b)) comparing country level recharge estimates from the current model with WaterGAP, and revised the discussion accordingly. We were not able to compare our results with PCR-GlobWB, as its country-wide recharge results are not publically available.

We added the following to Line 447-450: Recharge estimates from the best models in the present study were compared to recharge estimates from the complex hydrological model (WaterGAP) (Figure 11(b)). Even though the model in this study overestimates recharge for countries with fewer data points, the scatter shows a smaller spread compared to the FAO estimates

[Figure]

Figure 11. Comparison of predicted recharge against country level estimates from (a) FAO and (b) WaterGAP model.

**Line 455-467: While this paragraph discusses the influence of vegetation on recharge, the results fail to illustrate this influence. Please clarify how this influence was observed in the results.**

We have modified the discussion by highlighting the importance of vegetation as shown in our results.

Line 513-517: In this study Land Use (LU) was used as a proxy for vegetation. According to the results, LU was found to be one of the predictors having the highest Proportion of Evidence (PoE) (Figure 4). In addition, all the better performing models included LU as one of the predictors which clearly indicates that vegetation is one of the most influential factors for groundwater recharge.

**Line 486: Is this work able to say whether there are regions in the world which have declining or augmenting rates of recharge in the 1981-2014 time period?**

For addressing this comment, the following figures showing inter decadal percentage change in groundwater recharge are added in the supplementary material.

It is possible to say using the model whether the regions have declining or augmenting recharge rates. Hence the model is highly influenced by the changes in precipitation, the inter annual changes in the recharge will be highly correlated to that in precipitation.

[Figure]

[Figure]

Figure S1. Map showing change in mean percent decadal recharge (a) from 1981 to 2001 and (b) from 1991 to 2014. (Decadal change = mean decadal recharge of later decade – mean decadal recharge of former decade).

Marked up version of manuscript

[revised manuscript text omitted]